# A Multicenter Study about the Population Treated in the Respiratory Triage Stations Deployed by the Red Cross during the COVID-19 Pandemic

**DOI:** 10.3390/ijerph20010313

**Published:** 2022-12-25

**Authors:** José Antonio Ponce-Blandón, Rocío Romero-Castillo, Leyre Rodríguez-Leal, Raquel González-Hervías, Juan Francisco Velarde-García, Beatriz Álvarez-Embarba

**Affiliations:** 1Red Cross Nursing University Centre, University of Seville, 41009 Seville, Spain; 2International Federation of the Red Cross, Ecuador Headquarters, Quito 170403, Ecuador; 3Red Cross Nursing University College, Autonomous University of Madrid, 28003 Madrid, Spain; 4Research Group of Humanities and Qualitative Research in Health Science (Hum&QRinHS), Universidad Rey Juan Carlos, Avenida Atenas s/n, 28922 Alcorcon, Spain; 5Nursing Research Support Unit, Hospital General Universitario Gregorio Maranon, Calle Dr. Esquerdo 46, 28007 Madrid, Spain

**Keywords:** triage (DeCS; MeSH), coronavirus infections (DeCS; MeSH), respiratory tract diseases (DeCS; MeSH), health impact assessment (DeCS; MeSH)

## Abstract

Background: Care demand exceeded the availability of human and material resources during the COVID-19 pandemic, which is the reason why triage was fundamental. The objective is to know the clinical and sociodemographic factors of confirmed or suspected COVID-19 cases in triage stations from different Ecuadorian provinces. Method: A multicenter study with a retrospective and descriptive design. The patients included were those who accessed the Respiratory Triage stations deployed by the Ecuadorian Red Cross in eight Ecuadorian provinces during March and April 2021. Triage allows for selecting patients that need urgent treatment and favors efficacy of health resources. Results: The study population consisted of a total of 21,120 patients, of which 43.1% were men and 56.9% were women, with an age range between 0 and 98 years old. Severity of COVID-19 behaved differently according to gender, with mild symptoms predominating in women and severe or critical symptoms in men. Higher incidence of critical cases was observed in patients over 65 years old. It was observed that overweight predominated in critical, severe, and moderate cases, while the body mass index of patients with mild symptoms was within the normal range. Conclusions: The Ecuadorian Red Cross units identified some suspected COVID-19 cases, facilitating their follow-up and isolation. Fever was the most significant early finding.

## 1. Introduction

COVID-19 was identified in the city of Wuhan on 1 December 2019, due to an increase in the number of cases of pneumonia of unknown origin. The World Health Organization (WHO) declared it a health emergency on January 2020 and elevated it to the level of a world pandemic in March of the same year [1]. The care demand exceeded availability of human and material resources, which is why triage was fundamental. This is a process that allows managing clinical risk to adequately and safely handle the patient flows when the clinical demand and needs surpass the resources [2].

Triage allows selecting patients that need urgent treatment and favors efficacy of the health resources [3]. Some health systems have employed patient self-evaluation methods and symptom verifiers as a first contact point for the patients [4,5]. These tools have the potential to allocate resources, providing automated triage counseling and linking the patients to the optimum care level [6]. The basic values that underlie triage in crisis situations include prioritization of the medical urgency, which can be beneficial to the individual to provide an equal service [4].

During this COVID-19 pandemic, Ecuador is the second country in South America with the highest number of infections, with 22,719 confirmed cases of COVID-19, 576 deaths from COVID-19, and 1060 suspected deaths without confirmatory test of COVID-19 [7], so front-line health care assisted by specialists in infectious diseases has been fundamental. Individuals with COVID-19 can be asymptomatic and present some atypical manifestations [1]. The first step in most of the national and international protocols has been to rule out those patients that should not be considered as candidates for Intensive Care Units (ICUs) [8], as the number of available beds is limited, although it varies significantly across countries. Specifically, the total number of ICU beds in Ecuador was 2481 in 2020, which was insufficient for the overload faced by the country [9,10,11]. In addition, the nursing staff deficit has been associated with a marked increase in mortality, 64% more deaths than expected [12].

Owing to the scarcity of health professionals and material resources as well as to the increase in health costs, health systems have presented shocking social impacts during this pandemic. In many countries, investments in health have been insufficient to face the situation, as is the case of Ecuador. In this country, the Ecuadorian Red Cross (ERC) humanitarian help organization developed an intervention through triage stations [13].

The Ecuadorian deployments to control COVID-19 are relevant to the study considering its fragmented public health system and the heterogeneous evolution of the pandemic within different administrative units, i.e., its provinces [13,14]. For example, two of its main cities, Quito and Guayaquil, applied the recommended initial control measures at different moments. Guayaquil prohibited massive gatherings and implemented strict isolation approximately two weeks later than Quito [15]. The reports depended on the availability of resources and infrastructure, which generated a significant bias in the case count [15]. Adaptation strategies were also carried out in hospitals and patients based mainly on a new distribution of suspected patients, the creation of specific protocols, the use of triage, and the use of artificial intelligence in reading X-rays performed in triage [16].

This study was proposed due to the ongoing situation in Ecuador: the scarce number of studies investigating this issue, which are especially focused on patients with a confirmed diagnosis at a time when there was a shortage of diagnostic tests [17], and the important role played by the ERC organization in this pandemic with its triage stations, with the main objective of describing the clinical care profiles in the triage stations, together with knowing which clinical and sociodemographic factors can be related to the confirmed or suspected COVID-19 cases. The motivation and purpose of this study was to learn about the clinical profile of COVID-19 cases and the management of this pandemic situation. This can be useful for justifying new health measures or assessing ways to improve future situations.

## 2. Materials and Methods

A multicentral, observational, and retrospective study with a descriptive design was conducted. The patients included were all those that accessed the Respiratory Triage stations deployed by the ERC in the context of the agreements between this institution and the Ecuadorian Public Health Ministry (*Ministerio de Salud Pública del Ecuador*, *MSPE*) from 1 March 2021 to 30 April 2021, supported by the IFRC and funded by the USAID. 

In 8 provinces, a total of 22 health teams comprising a physician, a nurse, and a nursing assistant were deployed, all linked to areas attached to different health centers and hospitals and trained to perform respiratory triage activities, with the objective of mitigating the care burden of the MSPE public services. Appendix A reflects this distribution.

### 2.1. Triage Station Protocol

From reception of emergencies at the hospital and/or health center, all the patients that presented respiratory symptoms compatible with COVID-19 were referred to these triage stations, where they were subjected to the triage clinical protocol established by the MSPE 14 in the framework of the WHO guidelines for the management of COVID-19 [18,19] (see Appendix A).

The health team conducted a few sessions to sensitize the patients and/or companions, in order to remind them about the required prevention and isolation measures. Likewise, all the data for the ERC were recorded, and notification to the MSPE through Model 08 for the determination of confirmed and probable and/or suspected COVID-19 cases. This observational study was conducted in compliance with the Strengthening the Reporting of Observational Studies in Epidemiology (STROBE) statement [20].

Sociodemographic variables (age, gender, country or origin) and clinical/care variables (care center and province, date, presence of disability, Body Mass Index (BMI), temperature, heart rate, systolic/diastolic blood pressure, respiratory frequency, oxygen saturation, medical diagnosis based on the information collected from the Statistical International Classification of Diseases (ICD-10)) [21] were included and, in the case of confirmed or probable and/or suspected COVID-19 cases, the patients’ classification was based on their severity [18,19]. In addition, it was recorded whether guidance talks about preventive measures were offered. To apply the “U071 COVID-19, virus identified: CONFIRMED case with a POSITIVE test result”, it was indispensable to have a positive result in the RT-PCR test, whereas the “U072 COVID-19, virus not identified” diagnosis was recorded when the suspected and/or probable case criteria were met, established without the need for confirmation through a RT-PCR test.

### 2.2. Methods

The information source for the sociodemographic and care variables was the administrative care record through a form that was filled out via a KoBoToolBox (Kobo, MA, USA) link, which was accessed through a tablet device with an Internet connection. The teams were specifically trained for this record and its quality was monitored. 

For data analysis, some variables were recorded by grouping those categories that allowed facilitating the analysis without losing relevant information. Relative and absolute frequencies with 95% confidence intervals were calculated for the qualitative variables, as well as central tendency and dispersion measures for the continuous variables. The bivariate relationships between sociodemographic and care variables were analyzed by resorting to the chi-square test, acknowledging a statistically significant relationship when the *p*-value was below 0.05, and a multivariate analysis through step-by-step multinomial logistic regression on the “severity of the disease” dependent variable, taking as a reference the “moderate severity” category, where all the variables previously studied that had been significant were included as independent variables and possible factors. Data analysis was performed in the EpiInfo™ 7.2.4.0 (Centers for Disease Control and Prevention (CDC), Atlanta, GA, USA) and SPSS 20v^®^ programs (IBM, New York, NY, USA).

Authorization was obtained from the Research Ethics Committee of the University of Seville. The researchers had their access denied to all types of personal information. During the entire data collection process, the ethical principles for medical research in humans described in the latest revision of the Declaration of Helsinki were applied [22].

## 3. Results

The study population consisted of 21,120 patients, of which 43.1% were men ([42.4–43.8]; n = 9107) and 56.9% were women ([56.2–57.5]; n = 12,013). The age group corresponded to 0–98 years old, with a mean of 34.1 (SD = 20.02). 

Regarding the countries of origin, most of the patients were from Ecuador, representing 97.7% of the total of the sample treated ([97.4–97.8]; n = 20,630), followed by Venezuela with 1.5% ([1.4–1.7]; n = 331), and Colombia with 0.32% ([0.25–0.40]; n = 67); 99.1% of the patients treated did not present any acknowledged disability ([99.0–99.3]; n = 20,945). 

The triage stations with the largest volumes of patients treated were those from the provinces of Guayas, with 21.4% ([20.9–22.0]; n = 4527), Pichincha with 17.3% ([16.8–17.8]; n = 3667), and Tungurahua with 16.5% ([16.0–17.0]; n = 3494) (see Appendix A).

In relation to the clinical and care variables, it was observed that 11.5% of the patients attended the services with fever (temperature ≥ 37.8 °C) ([11.0–11.9]; n = 2274) and 10.9% with signs of hypoxia (O_2_ saturation ≤ 90%) ([10.5–11.3]; n = 2307); 96.4% of the patients treated were offered guidance talks ([96.1–96.6]; n = 20,362) to continue developing prevention and isolation behaviors.

Regarding the medical diagnoses, 13.4% of the patients ([12.9–13.9]; n = 2840) presented the “U071 COVID-19, virus identified: CONFIRMED case with a POSITIVE test result” diagnosis and the “U072 COVID-19, virus not identified” diagnosis corresponded to 29.9% ([29.3–30.6]; n = 6332). Another of the most frequent diagnoses was “J00X Acute rhinopharyngitis (common cold)”, which corresponded to 20.3% ([19.8–20.9]; n = 4310); 7.5% of the patients presented other additional diagnosis ([7.1–7.8]; n = 1601).

Based on the severity level, 74.7% of the patients categorized as “U071 COVID-19, virus identified” ([73.1–76.3]; n = 2078) needed clinical management for “mild” COVID-19. Clinical management for “severe” COVID-19 was necessary in 2.8% ([2.3–3.5]; n = 80), and 1.0% of these patients’ required management of the “critical” form of the disease ([0.7–1.5]; n = 30). For the patients categorized as “U072 COVID-19, virus not identified”, 84.9% ([84.0–85.7]; n = 5377) required clinical management for “mild” COVID-19, whereas 2.18% ([1.8–2.5]; n = 138) needed clinical management for “severe” COVID-19 and 0.22% ([0.1–0.3]; n = 14) needed it for the “critical” form of the disease. These results are presented in Table 1.

Arranged according to the sociodemographic variables, Table 2 and Table 3 show the bivariate analysis between the type of diagnosis and the severity of the symptoms. Regarding the type of diagnosis, statistically significant differences were found according to the province (*p* < 0.001), with Guayas and Pichincha being the ones with the most U071 and U072 diagnoses, respectively; according to sex (*p* < 0.001), with more suspected and confirmed diagnoses being made in women; according to age (*p* < 0.001) with U072 being more common in the group of 35–65 years. Differences were also observed between the groups according to BMI and guidance talk. An analysis of subgroups was conducted in the diagnoses, but the differences were not significant.

Severity of COVID-19 also behaves differently according to gender (*p* = 0.002); the mild form is the most common in both (women: 83.0%, n = 4202; men: 80.3%, n = 3250). However, in the mild and moderate forms, there is predominance of women (mild: 56.4%, n = 4202; moderate: 52.8%, n = 736), whereas the severe and critical forms were more frequent in men (severe: 52.5%, n = 114; critical: 56.8%, n = 25). 

In relation to age, although the number of patients aged over 65 years old treated in the Respiratory Triage stations was lower than in other groups (8.5%, n = 775), higher incidence of critical cases is observed in them: 2.2% (n = 17) versus 0.5% (n = 21) in the group from 36 to 65 years old and 0.2% in the group comprising individuals under the age of 35. The incidence of severe and moderate cases maintains this trend, being reversed in the mild cases, which represent 60.9% (n = 472) of the patients aged over 66 years old, 78.1% (n = 3402) in the group from 36 to 65 years old, and 90.0% (n = 3557) in the case of those under the age of 35, being statistically significant differences (*p* < 0.001). This trend is also maintained if we analyze the severity subgroups by age and gender (*p* < 0.001).

In relation to BMI and severity of COVID-19, it is noticed that overweight predominates in the critical (47.1%, n = 16), severe (41.5%, n = 49) and moderate (39.1%, n = 461) diagnoses, whereas the patients usually present normal weight (37.9%, n = 2071) or overweight (37.4%, n = 2048) in the mild diagnoses, being statistically significant differences (*p* < 0.001). If we also consider gender, it is noticed that men present a higher proportion of overweight in all the severity categories (critical: 50.0%, n = 10; severe: 44.4%, n = 28; moderate: 39.6%, n = 221; and mild: 37.9%, n = 933), whereas women with mild and critical symptoms present normal weight with 38.0% (n = 1144) and 42.9% (n = 6), respectively (*p* < 0.001).

Regarding the symptoms, in all the severity categories there is predominance of patients who attend the services with fever (*p* < 0.001), normal blood pressure (*p* < 0.001), and a heart rate between 60 and 120 bpm (*p* < 0.001). However, oxygen saturation worsens according to the severity level (*p* < 0.001), being below 90 in 77.5% (n = 31) of the critical patients and in 78.3% (n = 137) of the severe ones, and above 90 in 83.7% (n = 1122) of the moderate cases and in 95.5% (n = 6346) of the mild cases. 

Finally, a multinomial logistic regression was carried out to determine the weight of each symptom to predict the severity of COVID-19. Better fit and likelihood results were obtained with grouped age, BMI, fever, and saturation. Residential area, gender, blood pressure, heart rate, or respiratory frequency were not statistically significant and were excluded from the final analysis. As a result, the final model would explain between 14.8% and 21.7% of the differences observed, and only being significant for the patients classified as “mild” cases. Table 4 presents the final model with the parameters, their standard deviations (SD), the Wald statistical test, the Odds Ratios (OR), and the 95% confidence interval and the p-value. The model contains a total of 7783 individuals because it focuses on those suspected or confirmed cases of COVID-19 that do not contain missing values. The percentages of patients correctly classified in each group according to the model are 15.2% for severe, 5.2% for moderate and 98.8% for mild. It is important to highlight that there is an inversely proportional relationship in mild patients’ obesity and an oxygen saturation lower than 90.

## 4. Discussion

The results of this study are similar to those already published by Ortiz-Prado et al. [17] but it expands them with both confirmed and suspected of COVID-19 population. The concentration of cases detected in the Guayaquil area identified by the MSPE and by the Ecuadorian social observatory follows a similar behavior to the findings of this study: the province of Guayas with 21.4%, the province of Tungurahua with 16.5%, and the province of Pichincha with 17.4%. We notice that the most populated and best connected municipalities were affected earlier in time and that the least populated were affected in a later phase of the pandemic, similar effects to what happens in other South American countries such as Brazil [23]. 

Regarding distribution of the disease by gender, the data published by the MSPE differ from those found in this study; they indicate 51.55% incidence in men versus the value of 43.6% obtained in this paper, breaking invisibility of the gender differences that was perceived during the pandemic [24]. In this sense, our data are more similar to those published by some European countries, where more confirmed cases are also observed in women than in men [25]. For example, in Spain, initially it was more frequent in men, but as of March 31, the magnitude of the figures equaled as it increased in women. This same pattern has been observed in Belgium, Portugal, and the Netherlands. The explanation for this finding might be due to the care roles performed by women [26]. This gender bias is also observed in deaths, as indicated in the study by Cuéllar et al. [12], indigenous women in each age group have higher ED rates than the general population and, for ages 20–49, have higher ED rates than indigenous men. 

Likewise, the distribution of contagions by age found differs from the results obtained in other European countries. Incidence in the youngest patients might be explained by the impossibility of implementing Telework in Ecuador. In their study conducted in the New York City subway, Karla Therese et al. [27] showed that the SARS-CoV-2 exposure risk is higher in communities with low socioeconomic levels due to their more limited ability to stay in their homes and to the use of public transportation, which we call social distancing inequality. The study conducted by Defaz et al. [28] shows that the work activities most affected by COVID-19 are those related to agriculture, trade, and household chores, predominant activities in the city of Guayaquil [11,29]. 

It is considered as necessary to control and rigorously evaluate COVID-19 transmission, considering the social determinants, access to the health services, and delays in the diagnosis. Ecuador lacks sufficient capacity to perform the molecular diagnostic test (RT-PCR), which limits the epidemiological surveillance strategies and tracking of contacts, so that there is underdiagnosis of the disease, as in other Latin American countries where the data reveal a socioeconomic bias in the tests and diagnoses [27], which is reflected in the high lethality rate (4.9%), considerably above the values found in other countries. 

The ERC triage units have failed to perform more diagnostic tests, although they have in fact identified more suspected cases, thus facilitating their follow-up and isolation and training in COVID-19 prevention. Although most of the cases are mild, an increase in other more severe categories was indeed noticed in April; these data coincide with the increase in the number of deaths and with the epidemiological situation recorded by the Ministry: 624 people with a reserved prognosis and 2046 stable individuals; therefore, the triage units were able to identify a higher number of cases, thus allowing for social isolation and follow-up of mild and moderate cases, as well as for referral of the severe and critical cases. 

Regarding the symptoms observed, fever has become the most significant early finding when it comes to identifying suspected cases, to the extent of devising fever clinics specialized in classifying them and tracing the patients’ flow within the triage systems, in relation to the epidemiological history; even assigning them a different color code. In their studies, Huang et al. [30] classified patients with fever and no epidemiological history with a green QR code versus those who did have such history (red code), separating and allocating them to the corresponding treatments. The fever clinics were initially developed in China, as epicenter of the pandemic, in order to mitigate overload in hospitals and prevent the risk of cross-infections [31], especially through the Teleconsultation service. In addition to fever, Li et al. [31] described coughing, pharyngalgia, headache, rhinorrhea, expectoration, abdominal pain, and chest distress; however, they did not describe desaturation or changes in heart rate, respiratory frequency, or blood pressure. For being a telephone counseling system, it was not possible to monitor saturation of the patients, unlike in our study.

Wang et al. [3] conducted a similar study on the triage of patients treated at the Central Hospital of Xi’an (China) with 25,742 subjects, of which 246 were diagnosed with COVID-19; the detection rate for suspected cases was 1.63% (4 out of 246), requiring more conclusive tests such as blood analysis exams and CAT scans according to their epidemiological history in relation to exposure to the virus. Fever was the main cause for investigation, with predominance of unknown origin followed by fever arising from upper tract infections and due to pneumonias. Likewise, the same similarities regarding onset of fever can be seen, without being conclusive as to its origin given the absence of confirmatory evidence.

One of the main limitations is the retrospective nature of this study, which has not allowed us to include clinical and demographic variables that are currently known to influence the development of the disease. However, the information was collected following the same criteria owing to the fact that the professionals received training prior to the opening of the units.

Another limitation is the time in which it was conducted, circumscribed only to the third wave of COVID-19 in Ecuador through the triage stations created by CRE, but it is consistent with the previously consulted bibliography.

## 5. Conclusions

The ERC triage units identified a large number of suspected COVID-19 cases, facilitating their follow-up and isolation. Fever was the most significant early finding. Thanks to the classification and organization implemented, it was possible to alleviate saturation of the health services and to prevent infections.

Relevance to clinical practice:

The present study has allowed us to know the sociodemographic characteristics of the third wave of COVID-19 in Ecuador through the triage stations created by CRE and highlights the activity carried out by nurses in these triage units.

## Figures and Tables

**Table 1 ijerph-20-00313-t001:** Descriptive analysis of the clinical and care variables.

Fever	n	%	95% CI
No: <37.8 °C	17,504	88.5%	[88.0–88.9]
Yes: ≥37.8 °C	2274	11.5%	[11.0–11.9]
Total	19,778	100%	
O_2_ saturation	n	%	95% CI
O_2_ saturation ≤ 90%	2307	10.9%	[10.5–11.3]
O_2_ saturation > 90%	18,813	89.1%	[88.6–89.5]
Total	21,120	100%	
BMI category	n	%	95% CI
Underweight < 18.5	2028	12.2%	[11.7–12.7]
Normal weight [18.5–24.9]	5756	34.6%	[33.9–35.3]
Overweight [25–29.9]	5662	34.1%	[33.3–34.8]
Obesity ≥ 30	3164	19.0%	[18.4–19.6]
Total	16,610	100%	
Parameter	Mean, standard deviation and range		
Systolic blood pressure	117.7 mmHg (SD = 16.3) [60–230 mmHg]		
Diastolic blood pressure	73.7 mmHg (SD = 10.5) [40–130 mmHg]		
Heart rate	89.3 bpm (SD = 18.1) [35–125 bpm]		
Respiratory frequency	20.8 brpm (SD = 6.6) [10–110 brpm]		
Guidance talks offered	n	%	95% CI
Yes	20,362	96.4%	[96.1–96.6]
No	758	3.59%	[3.3–3.8]
Patient’s ICD-10 diagnosis:	n	%	95% CI
U072 COVID-19, virus not identified	6332	30.0	[29.30–30.6]
J00X Acute rhinopharyngitis (common cold)	4301	20.4	[19.8–20.9]
U071 COVID-19, virus identified: CONFIRMED case with a POSITIVE test result.	2840	13.4	[13.0–13.9]
J029 Acute pharyngitis, unspecified	2214	10.5	[10.0–10.9]
J039 Acute tonsillitis, unspecified	1627	7.7	[7.3–8.0]
Other ICD-10 diagnoses not related to the airways	1601	7.6	[7.2–7.9]
Other ICD-10 diagnoses related to the airways	1055	5.0	[4.7–5.3]
J02 Acute pharyngitis	501	2.4	[2.2–2.6]
J069 Acute infection of the upper airways, unspecified	226	1.1	[0.9–1.2]
J030 Streptococcus tonsillitis	222	1.1	[0.9–1.2]
J03 Acute tonsillitis	201	1.0	[0.8–1.1]
Total	21,120	100	
U071 Covid-19 Severity Level	n	%	95% CI
Management for mild COVID-19	2078	74.7%	[73.1–76.3]
Management for moderate COVID-19	591	21.2%	[19.8–22.8]
Management for severe COVID-19	80	2.8%	[2.3–3.5]
Management for critical COVID-19	30	1.0%	[0.7–1.5]
Total	2779	100%	
U072 Covid-19 Severity Level	n	%	95% CI
Management for mild COVID-19	5377	84.9%	[84.0–85.7]
Management for moderate COVID-19	803	12.6%	[11.8–13.5]
Management for severe COVID-19	138	2.1%	[1.8–2.5]
Management for critical COVID-19	14	0.2%	[0.1–0.3]
Total	6332	100%	

**Table 2 ijerph-20-00313-t002:** Type of diagnosis and severity level according to sociodemographic characteristics.

		COVID-19 Diagnosis			
		U071	U072	Others	Chi-Square
		N = 2840	N = 6332	N = 11,948	(*p*-Value)
		n (%)	n (%)	n (%)	
**Province**					<0.001
	Azuay	84 (14.1)	181 (30.4)	330 (55.5)	
	El Oro	61 (3.2)	784 (40.6)	1084 (56.2)	
	Guayas	1231 (27.2)	794 (17.5)	2502 (55.3)	
	Los Ríos	204 (13.9)	500 (34.1)	761 (51.9)	
	Manabí	198 (8.8)	572 (25.4)	1484 (65.8)	
	Pichincha	333 (9.1)	1691 (46.1)	1643 (44.8)	
	Santo Domingo	570 (17.9)	1080 (33.9)	1539 (48.3)	
	Tungurahua	159 (4.6)	730 (20.9)	0.2605 (74.6)	
**Gender**					21.475 (<0.001)
	Male	1332 (14.6)	2742 (30.1)	5033 (55.3)	
	Female	1508 (12.6)	3590 (29.9)	6915 (57.6)	
**Age**					1061.753 (<0.001)
	≤35 years old	1014 (9.3)	2959 (27.3)	6681 (63.4)	
	35–65 years old	1502 (18.2)	2890 (35.1)	3850 (46.7)	
	≥65 years old	324 (22.1)	459 (31.3)	683 (46.6)	
**BMI**					1264.138 (<0.001)
	Underweight	54 (2.7)	171 (8.4)	1803 (88.9)	
	Normal	784 (13.6)	1660 (28.8)	3312 (57.7)	
	Overweight	917 (16.2)	1691 (29.9)	3054 (53.9)	
	Obese	612 (19.3)	975 (30.8)	1577 (49.8)	
**Guidance talk**					184.265 (<0.001)
	No	36 (4.7)	112 (14.8)	610 (80.5)	
	Yes	2804 (13.8)	6220 (30.5)	11,338 (55.7)	

The percentages calculated are for the distribution of groups according to rows.

**Table 3 ijerph-20-00313-t003:** Severity level according to sociodemographic characteristics.

		COVID-19 Severity				
		Critical	Severe	Moderate	Mild	Chi-Square
		N = 44	N = 217	N = 1393	N = 7452	(*p*-Value)
		n (%)	n (%)	n (%)	n (%)	
**Province**						517.4 (<0.001)
	Azuay	2 (0.8)	38 (14.3)	37 (14.0)	188 (70.9)	
	El Oro	3 (0.4)	19 (2.3)	183 (21.7)	637 (75.7)	
	Guayas	17 (0.8)	18 (0.9)	507 (25.1)	1477 (73.2)	
	Los Ríos	1 (0.1)	23 (3.3)	119 (17.1)	552 (79.4)	
	Manabí	0 (0)	2 (0.3)	71 (9.2)	695 (90.5)	
	Pichincha	9 (0.4)	72 (3.6)	181 (9.0)	1755 (87.0)	
	Santo Domingo	11 (0.7)	34 (2.1)	203 (12.4)	1386 (84.8)	
	Tungurahua	1 (0.1)	11 (1.3)	92 (10.6)	762 (88.0)	
**Gender**						14.7 (0.002)
	Male (n = 4046)	25 (0.6)	114 (2.8)	657 (16.2)	3250 (80.3)	
	Female (n = 5056)	19 (0.4)	217 (2.4)	736 (14.5)	4202 (83.0)	
**Age**						533.5 (<0.001)
	≤35 years old (n = 4354)	6 (0.2)	29 (0.7)	361 (9.1)	3557 (90.0)	
	35–65 years old (n = 3953)	21 (0.5)	119 (2.7)	812 (18.6)	3402 (78.1)	
	≥65 years old (n = 775)	17 (2.2)	69 (8.9)	217 (28.0)	472 (60.9)	
**BMI**						96.3 (<0.001)
	Underweight (n = 224)	0 (0.0)	1 (0.4)	26 (11.6)	197 (87.9)	
	Normal (n = 2433)	12 (0.5)	34 (1.4)	316 (13.0)	2071 (85.1)	
	Overweight (n = 2574)	16 (0.6)	49 (1.9)	461 (17.9)	2048 (79.6)	
	Obese (n = 1568)	6 (0.4)	34 (2.2)	375 (23.9)	1156 (73.5)	

The percentages calculated are for the distribution of groups according to rows.

**Table 4 ijerph-20-00313-t004:** Final multinomial logistic regression model with the “moderate” category as a reference.

Severity	Independent Variables		B (SD)	Wald Test	OR (95% CI)	*p*-Value
Critical (40; 0.5%)	Fever	No	−0.473 (0.399)	1.40	0.62 (0.28–1.36)	0.236
	BMI	Underweight	−18.873 (0.0)		6.36^−9^ (6.36 × 10^−9^–6.36× 10^−9^)	<0.001
		Normal	0.286 (0.403)	0.50	1.33 (0.60–2.93)	0.478
		Obese	−0.578 (0.495)	1.36	0.56 (0.21–1.48)	0.243
	Age	≤35	−0.630 (0.549)	1.32	0.53 (0.18–1.56)	0.251
		(36, 65)	−0.549 (0.364)	2.27	0.58 (0.18–1.56)	0.131
	O_2_ saturation	≤90	2.933 (0.406)	52.12	18.79 (8.47–41.67)	<0.001
Severe (171; 2.2%)	Fever	No	−1.068 (0.198)	29.21	0.34 (0.23–0.51)	<0.001
	BMI	Underweight	−1.198 (1.088)	1.83	1.38 (0.86–2.21)	0.271
		Normal	0.162 (0.256)	0.40	1.18 (0.71–1.94)	0.526
		Obese	0 (0.254)	0.0	1.00 (0.60–1.64)	0.999
	Age	≤35	−0.462 (0.284)	2.63	0.63 (0.36–1.10)	0.105
		(36, 65)	−0.383 (0.207)	3.42	0.68 (0.45–1.02)	0.064
	O_2_ saturation	≤90	2.796 (0.214)	170.09	16.37 (10.76–24.92)	<0.001
Mild (6276; 80.6%)	Fever	No	0.434 (0.087)	25.08	1.54 (1.30–1.83)	<0.001
	BMI	Underweight	0.212 (0.228)	0.87	1.24 (0.79–1.93)	0.352
		Normal	0.247 (0.082)	8.98	1.28 (1.09–1.50)	0.003
		Obese	−0.379 (0.082)	21.52	0.68 (0.58–0.803)	<0.001
	Age	≤35	1.260 (0.106)	140.77	3.526 (2.86–4.34)	<0.001
		(36, 65)	0.605 (0.098)	38.13	1.83 (1.51–2.22)	<0.001
	O_2_ saturation	≤90	−1.492 (0.107)	195.99	0.22 (0.18–0.28)	<0.001

## Data Availability

Not applicable.

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
