# Peer review of "A Multicenter Study about the Population Treated in the Respiratory Triage Stations Deployed by the Red Cross during the COVID-19 Pandemic"

_ijerph, 2022, doi:10.3390/ijerph20010313_

Round 1

Reviewer 1 Report (Previous Reviewer 1)

Please see the comments attached.

Author Response

Report on ijerph-2084392

This paper aims to describe the clinical and social-economic factors of the population that entered triage stations to treat suspected COVID-19 cases. The authors found that older, overweight, and male patients tended to experience more severe symptoms. The paper was originally submitted and reviewed in October 2022. My comments at that timewere mainly concerned with the background information, the contribution of the paper, and the description of the models estimated. I appreciate that the authors have answered all of my questions in the cover letter and addressed some of my comments in the manuscript.

Thank you very much for reviewing our manuscript again, we have worked to include all the proposed comments and we appreciate that you take into account all the work done since our first submission.

Major comments

Although you wrote in the cover letter that the limited space of the article does not allow them to present the knowledge from the previous papers, and they have added citations to that section, I still think demonstrating how your paper fills the gap in the literature is one of the most important things for an academic paper, and that information usually does not take more than a paragraph. Based on your answer in the cover letter, I suggest that you add one or two sentences saying that there is/are no/two papers looking at this issue, and then you explain how your paper is different from them. You probably look at different types of healthcare for Covid patients (such as ad-hoc rather than more sustainable formats), look at different characteristics of patients, or use a different dataset.

Thank you very much for highlighting that gap. We have performed a new pubmed search to include recently published articles on our topic. The search "COVID-19"[Mesh] AND "Triage"[Mesh] AND "Ecuador"[Mesh] provides 3 articles of which one has been selected, expanding the introduction with the following paragraph on the hospital adaptations made:

“Adaptation strategies were also carried out in hospitals and patients based mainly on a new distribution of suspected patients, the creation of specific protocols, the use of triage, and the use of artificial intelligence in reading X-rays performed in triage [16].”

Another more general search performed ("COVID-19"[Mesh] AND "Ecuador"[Mesh]) returns 107 of which only one article is similar to ours, focusing on the epidemiology, symptomatology, and mortality of confirmed cases during the first months. of pandemic. However, it should be noted that our research focuses on both suspected and confirmed cases who enter the hospital through respiratory triage, thus expanding the study population. This has been highlighted in the introduction by adding:

“the scarce number of studies looking at this issue, which are especially focused on pa-tients with a confirmed diagnosis at a time when there was a shortage of diagnostic tests [17]”

And, also, at the discussion:

“The results of this study are similar to those already published by Ortiz-Prado et al. [17] but it expands them with both confirmed and suspected of Covid-19 population.”

In Section 2. Materials and Methods, you added subsection 2.1. Triage Station Protocols, which aims to separate the program description and the method. However, there is no subsection 2.2. I suggest that you add Section 2.2. Methods, which starts from line 113.

Thank you very much for this comment, we have added the new section.

And finally, I’d still like to know what variables were added in the previous step (bivariate regressions) but were not significant and therefore were not presented in the paper. I understand that the authors are hesitant that it would increase the length of the paper, but it isimportant to summarize those results for both transparency and information purposes. Informing the readers/policy-makers what is statistically insignificant is no less important than informing them what is significant. Besides, it takes one sentence to include that information, for example, “differences in residential area and age are not significant and therefore are not presented in the paper.”

Thank you very much for your comment, we have been able to improve this part of the results by adding:

“Residential area, gender, blood pressure, heart rate or respiratory frequency were not statis-tically significant and were excluded of in the final analysis.”

Minor comments

There are some grammar errors in the paper.

The English language of the article has been reviewed and certified by Latintrad.

This manuscript is a resubmission of an earlier submission. The following is a list of the peer review reports and author responses from that submission.

Round 1

Reviewer 1 Report

Report on ijerph-1987946

Ecuador deployed triage stations to treat suspected COVID-19 cases due to a lack of health resources. This paper aims to describe the clinical and social-economical factors of the population that entered those stations.  The authors found that older, overweight, and male patients tended to experience more severe symptoms.  

I enjoyed reading the details about the characteristics of the patients described in the paper. I have the following comments.

Major comments

First, my main concerns are the contribution the paper makes compared to what we know about the demographic factors of COVID-19 patients, and how the findings contribute to improving the specific COCID-19 situation in Ecuador. To demonstrate the significance of the paper, the authors need to discuss previous literature on the characteristics of the population of COVID-19 patients in the world, why we need to know the demographic factors of Ecuador’s patients specifically, and whether other papers have described the population of COVID 19 patients in Ecuador. The authors also need to discuss how the findings can be applied to policies or clinical practice in Ecuador.

Second, the authors currently describe the program, including locations of triage centers and the procedure used to admit patients into those centers, in “Section 2. Materials and Methods.” I suggest that you describe the program in a separate section.

Third, although the authors wrote that sociodemographic and care information was recorded through a form on the internet, it was not clear whether the data were administrative (officially collected by care providers) or survey data.

Fourth, the authors mentioned that they explored the bivariate relationship between socioeconomic factors and care variables and included the factors that have statistically significant relationships in multivariate models. Please describe all the variables that you explore their bivariate relationship in the first stage.

And finally, I think it would be interesting if the authors could provide information about patient characteristics stratified by treatment results, so health professionals and policymakers could see the population at risk.

Minor comments

In the Discussion section, the authors mentioned that the distribution of contagion by age differed from findings from European countries. Please be more specific about the differences and add citations so that readers know what research they used for this comparison.

There are typos and grammar errors here and there in the paper.

Reviewer 2 Report

Comments on "A multicenter study about the population treated in the respiratory triage situations deployed by the red cross during the COVID-19 pandemic"

This article addresses an interesting topic: the characterization of the population treated in the triage stations offered by the Red Cross during the COVID-19 pandemic in Ecuador. There are, however, some points to be considered. Please find some comments and suggestions below:

Major points:

  • The motivation of the paper should be better developed. The intention of the authors to identify the clinical and social factors related to the suspected/confirmed cases of COVID-19 is perceived. Still, in some parts of the paper, there is a lack of a guiding line, which demotivates the reader. For example, what would the results be for? Would they help to draw conclusions that support the adoption of health policies that target specific groups? Would they reinforce the importance of NGOs in countries with less structured health systems, not just for the reality of COVID-19? I would like to see these issues addressed in the introduction or discussion, even if not exhaustively. In addition, I think the paper would benefit from an additional paragraph or two explaining the context of the Ecuadorian Health System in the Introduction section.
  • Table 4 shows the p-values associated with the Chi-square statistics for the Chi-square test of independence. On page 7, the authors begin the analysis of Table 4 with "Statistically significant differences were found (p<0.001)…" without noting that they are already analyzing a different table and without introducing the Chi-square test of independence. In fact, there should be a better transition in the analysis of the various tables. Otherwise, it is difficult to read and grab readers' attention.
  • Table 4 raises additional questions for me. The percentages add up vertically in the previous tables, while they add up horizontally in Table 4 without a note explaining this. Also, I think there are errors in the analysis. For instance, the variables gender, age and BMI all have a p-value <0.001, so the sentence on page 7, "The differences between diagnoses, provinces and genders were not significant," is incorrect, or this sentence "In relation to gender, women attended the Triage stations more frequently (p<0.001); they represented 56.9% (n=12,013) of the assistance appointments versus 43.1% (n=9,107) in men" does not come from Table 4. In short, the authors should reformulate the tables' analysis both to correct errors and simplify the writing to make it easier to read.
  • I have several considerations to make regarding the estimation of the multinomial logistic regression. The paper should present the model's motivation more carefully; that is, the multinomial logit serves to answer which specific questions? Why is the gender variable not included in the model? The results from Table 5 should also be better interpreted, more specifically with an additional estimation of the marginal effects. Furthermore, the number of observations substantially reduces from 21,120 to 8,095 patients. The authors recognize that this reduction is due to missing values. I think it would be more accurate to say that this analysis only focuses on suspected or confirmed cases of COVID-19, which explains the reduction in the number of observations.
  • What is the point of estimating a multinomial logistic model when the results of that model are not addressed in the Discussion section?

Minor points:

  • Please write the full term the first time you use ERC in the main text. In the abstract, use the full term.
  • Consider moving some tables or figures to the Appendix, more specifically, Table 1, Figure 1, and Table 2.
  • What was statistical software (and version) used in the statistical analysis? The authors should include this information in the paper.

Round 2

Reviewer 2 Report

Comments on “A multicenter study about the population treated in the respiratory triage stations deployed by the Red Cross during the COVID-19 pandemic”

I should note some improvements in the paper, in line with what was suggested. 

However, I am not very comfortable with ​​a paper that estimates a multinomial logistic model without interpreting the estimated coefficients or presenting the “average marginal effects”. Without this, the readers can extract little information from the estimation. 

The authors argue that “for reasons of space in the article and based on the results obtained in the model, we consider it a priority to expose the table and not its explanation, since as we say, only some variables enter into it, which supports that only explain between 14.8% and 21.7% of the observed differences”. However, I must say that an analysis based exclusively on R2 is very limited. Therefore, if the authors do not intend to interpret the results of the multinomial model, I think they should consider removing it from the paper.

Author Response

Thank you very much for your comments. In line with their recommendations, we have decided to eliminate the multivariate analysis so as not to confuse the reader. We have also improved some errors in the English translation of the document.